# Online Meta-Critic Learning for Off-Policy Actor-Critic Methods

**Wei Zhou**[*1]**, Yiying Li**[*1]**, Yongxin Yang**[2]**, Huaimin Wang**[1]**, Timothy M. Hospedales**[2,3]
[1]College of Computer, National University of Defense Technology
[2]School of Informatics, The University of Edinburgh
[3]Samsung AI Centre, Cambridge
{zhouwei14, liyiying10, hmwang}@nudt.edu.cn; {yongxin.yang, t.hospedales}@ed.ac.uk

## Abstract

Off-Policy Actor-Critic (OffP-AC) methods have proven successful in a variety of continuous control tasks. Normally, the critic's action-value function is updated using temporal-difference, and the critic in turn provides a loss for the actor that trains it to take actions with higher expected return. In this paper, we introduce a flexible *meta*-critic framework based on observing the learning process and meta-learning an additional loss for the actor that accelerates and improves actor-critic learning. Compared to existing meta-learning algorithms, meta-critic is rapidly learned online for a single task, rather than slowly over a family of tasks. Crucially, our meta-critic is designed for off-policy based learners, which currently provide state-of-the-art reinforcement learning sample efficiency. We demonstrate that online meta-critic learning benefits to a variety of continuous control tasks when combined with contemporary OffP-AC methods DDPG, TD3 and SAC.

## 1 Introduction

Off-policy Actor-Critic (OffP-AC) methods are currently central in deep reinforcement learning (RL) research due to their greater sample efficiency compared to on-policy alternatives. On-policy learning requires new trajectories to be collected for each update to the policy, and is expensive as the number of gradient steps and samples per step increases with task-complexity even for contemporary TRPO [33], PPO [34] and A3C [27] algorithms.

Off-policy methods, such as DDPG [20], TD3 [9] and SAC [13] achieve greater sample efficiency as they can learn from randomly sampled historical transitions without a time sequence requirement, making better use of past experience. The critic estimates action-value (Q-value) function using a differentiable function approximator, and the actor updates its policy parameters in the direction of the approximate action-value gradient. Briefly, the critic provides a loss to guide the actor, and is trained in turn to estimate the environmental action-value under the current policy via temporal-difference learning [38]. In all these cases the learning objective function is hand-crafted and fixed.

Recently, meta-learning [14] has become topical as a paradigm to accelerate RL by learning aspects of the learning strategy, for example, learning fast adaptation strategies [7, 30, 31], losses [3, 15, 17, 36], optimisation strategies [6], exploration strategies [11], hyperparameters [40, 42], and intrinsic rewards [44]. However, most of these works perform meta-learning on a family of tasks or environments and amortize this huge cost by deploying the trained strategy for fast learning on a new task.

---

[*]Contributed equally.

In this paper we introduce a meta-critic network to enhance OffP-AC learning methods. The meta-critic augments the vanilla critic to provide an additional loss to guide the actor's learning. However, compared to the vanilla critic, the meta-critic is explicitly (meta)-trained to accelerate the learning process rather than merely estimate the action-value function. Overall, the actor is trained by both critic and meta-critic provided losses, the critic is trained by temporal-difference as usual, and crucially the meta-critic is trained to generate maximum learning progress in the actor. Both the critic and meta-critic use randomly sampled transitions for effective OffP-AC learning, providing superior sample efficiency compared to existing on-policy meta-learners. We emphasize that meta-critic can be successfully learned *online* within a single task. This is in contrast to the currently widely used meta-learning paradigm – where entire task *families* are required to provide enough data for meta-learning, and to provide new tasks to amortize the huge cost of meta-learning.

Our framework augments vanilla AC learning with an additional meta-learned critic, which can be seen as providing intrinsic motivation towards optimum actor learning progress [28]. As analogously observed in recent meta-learning studies [8], our loss-learning can be formalized as bi-level optimisation with the upper level being meta-critic learning, and lower level being conventional learning. We solve this joint optimisation by iteratively updating the meta-critic and base learner online in a single task. Our strategy is related to the meta-loss learning in EPG [15], but learned online rather than offline, and integrated with OffP-AC rather than their on-policy policy-gradient learning. The most related prior work is LIRPG [44], which meta-learns an intrinsic reward online. However, their intrinsic reward just provides a helpful scalar offset to the environmental reward for on-policy trajectory optimisation via policy-gradient [37]. In contrast our meta-critic provides a loss for direct actor optimisation using sampled transitions, and achieves dramatically better sample efficiency than LIRPG reward learning. We evaluate several continuous control benchmarks and show that online meta-critic learning can improve contemporary OffP-AC algorithms including DDPG, TD3 and SAC.

## 2 Background and Related Work

**Policy-Gradient (PG) RL Methods.** Reinforcement learning involves an agent interacting with environment $E$. At each time $t$, the agent receives an observation $s_t$, takes a (possibly stochastic) action $a_t$ based on its policy $\pi : \mathcal{S} \to \mathcal{A}$, and receives a reward $r_t$ and new state $s_{t+1}$. The tuple $(s_t, a_t, r_t, s_{t+1})$ describes a state transition. The objective of RL is to find the optimal policy $\pi_\phi$, which maximizes the expected cumulative return $J$.

In on-policy RL, $J$ is defined as the discounted episodic return based on a sequential trajectory over horizon $H$: $(s_0, a_0, r_0, s_1 \cdots, s_H, a_H, r_H, s_{H+1})$. $J = \mathbb{E}_{r_t, s_t \sim E, a_t \sim \pi} \left[ \sum_{t=0}^{H} \gamma^t r_t \right]$. In on-policy AC, $r$ is represented by a surrogate state-value $V(s_t)$ from its critic. Since $J$ is only a scalar value that is not differentiable, the gradient of $J$ with respect to policy $\pi_\phi$ has to be optimised under the policy gradient theorem [37]: $\nabla_\phi J(\phi) = \mathbb{E}\left[ J \nabla_\phi \log \pi_\phi(a_t|s_t) \right]$. However, with respect to sample efficiency, even exploiting tricks like importance sampling and improved application of A2C [44], the use of full trajectories is less effective than the use of individual transitions by off-policy methods.

Off-policy actor-critic architectures provide better sample efficiency by reusing past experience (previously collected transitions). DDPG [20] borrows two main ideas from Deep Q Networks [25, 26]: a replay buffer and a target Q network to give consistent targets during temporal-difference backups. TD3 (Twin Delayed Deep Deterministic policy gradient) [9] develops a variant of Double Q-learning by taking the minimum value between a pair of critics to limit over-estimation, and the computational cost is reduced by using a single actor optimised with respect to $Q_{\theta_1}$. SAC (Soft Actor-Critic) [12, 13] proposes a maximum entropy RL framework where its stochastic actor aims to simultaneously maximize expected action-value and entropy. The latest version of SAC [13] also includes the "the minimum value between both critics" idea in its implementation. Specifically, in these off-policy AC methods, parameterized policies $\pi_\phi$ can be directly updated by defining actor loss in terms of the expected return $J(\phi)$ and taking its gradient $\nabla_\phi J(\phi)$, where $J(\phi)$ depends on the action-value $Q_\theta(s, a)$. Based on a batch of transitions randomly sampled from the buffer, the loss for actor provided by the critic is basically calculated as:

$$L^{\text{critic}} = -J(\phi) = -\mathbb{E}_{s \sim p_\pi} Q_\theta(s, a)|_{a = \pi_\phi(s)}. \tag{1}$$

Specifically, the loss $L^{\text{critic}}$ for actor in TD3 and SAC is calculated as Eq. (2) and Eq. (3) respectively:

$$L_{\text{TD3}}^{\text{critic}} = -\mathbb{E}_{s \sim p_\pi} Q_{\theta_1}(s, a)|_{a = \pi_\phi(s)}; \tag{2}$$

$$L_{\text{SAC}}^{\text{critic}} = E_{s \sim p_\pi}[\alpha \log(\pi_\phi(a|s)) - Q_\theta(s, a)|_{a = \pi_\phi(s)}]. \tag{3}$$

The actor is then updated as $\Delta\phi = \alpha\nabla_\phi L^{\text{critic}}$, following the critic's gradient to increase the likelihood of actions that achieve a higher Q-value. Meanwhile, the critic $\theta$ uses Q-learning updates to estimate the action-value function:

$$\theta \leftarrow \underset{\theta}{\arg\min} \ \mathbb{E}(Q_\theta(s_t, a_t) - r_t - \gamma Q_\theta(s_{t+1}, \pi(s_{t+1})))^2. \tag{4}$$

**Meta Learning for RL.** Meta-learning (a.k.a. learning to learn) [7, 14, 32] has received a resurgence in interest recently due to its potential to improve learning performance and sample efficiency in RL [11]. Several studies learn optimisers that provide policy updates with respect to known loss or reward functions [1, 6, 23]. A few studies learn hyperparameters [40, 42], loss functions [3, 15, 36] or rewards [44] that steer the learning of standard optimisers. Our meta-critic framework is in the category of loss-function meta-learning, but unlike most of these we are able to meta-learn the loss function online in parallel to learning a single extrinsic task rather. No costly offline learning on a task family is required as in Houthooft et al. [15], Sung et al. [36]. Most current Meta-RL methods are based on on-policy policy-gradient, limiting sample efficiency. For example, while LIRPG [44] is one of the few prior works to attempt online meta-learning, it is ineffective in practice due to only providing a scalar reward increment rather than a loss for direct optimisation. A few meta-RL studies have begun to address off-policy RL, for conventional multi-task meta-learning [30] and for optimising transfer vs forgetting in continual learning of multiple tasks [31]. The contribution of our Meta-Critic is to enhance state-of-the-art single-task OffP-AC RL with online meta-learning.

**Loss Learning.** Loss learning has been exploited in 'learning to teach' [41] and surrogate loss learning [10, 16] where a teacher network predicts the parameters of a manually designed loss in the supervised learning. In contrast our meta-critic is itself a differentiable loss, and is designed for use in RL. Other applications learn losses that improve model robustness to out of distribution samples [2, 19]. Some recent loss learning studies in RL focus mainly on the multi-task adaptation scenarios [3, 15, 36] or the generalization to entirely different environments [17]. Our loss learning architecture is related to Li et al. [19], but designed for accelerating single-task OffP-AC RL rather than improving robustness in multi-domain supervised learning.

## 3 Methodology

We aim to learn a meta-critic which augments the vanilla critic by providing an additional loss $L_\omega^{\text{mcritic}}$ for the actor. The vanilla loss for the policy (actor) is $L^{\text{critic}}$ given by the conventional critic. The actor is trained by $L^{\text{critic}}$ and $L_\omega^{\text{mcritic}}$ via stochastic gradient descent. The meta-critic parameter $\omega$ is optimized by meta-learning to accelerate actor learning progress. Here we follow the notation in TD3 and SAC that $\phi$ and $\theta$ denote actors and critics respectively.

**Algorithm Overview.** We train a meta-critic loss $L_\omega^{\text{mcritic}}$ that augments the vanilla critic $L^{\text{critic}}$ to enhance actor learning. Specifically, it should lead to the actor $\phi$ having improved performance on the normal task, as measured by $L^{\text{critic}}$ on the validation data, *after* learning on both meta-critic and vanilla critic losses. This can be seen as a bi-level optimisation problem[1] [8, 14, 29] of the form:

$$\omega = \underset{\omega}{\arg\min} \ L^{\text{meta}}(d_{val}; \phi^*)$$
$$s.t. \ \phi^* = \underset{\phi}{\arg\min} \ (L^{\text{critic}}(d_{trn}; \phi) + L_\omega^{\text{mcritic}}(d_{trn}; \phi)), \tag{5}$$

where we can assume $L^{\text{meta}}(\cdot) = L^{\text{critic}}(\cdot)$ for now. $d_{trn}$ and $d_{val}$ are different transition batches from replay buffer. Here the lower-level optimisation trains actor $\phi$ to minimize both the normal loss and meta-critic-provided loss on training samples. The upper-level optimisation further requires meta-critic $\omega$ to have produced a learned actor $\phi^*$ that minimizes a meta-loss that measures actor's normal performance on a set of validation samples, *after being trained by meta-critic*. Note that in principle the lower-level optimisation could purely rely on $L_\omega^{\text{mcritic}}$ analogously to the procedure in EPG [15], but we find optimising their sum greatly increases learning stability and speed. Eq. (5) is satisfied when meta-critic successfully trains the actor for good performance on the normal task

**Algorithm 1** Online Meta-Critic Learning for OffP-AC RL

$\phi, \theta, \omega, \mathcal{D} \leftarrow \emptyset$          // Initialise actor, critic, meta-critic and buffer
**for** each iteration **do**
   **for** each environment step **do**
      $a_t \sim \pi_\phi(a_t|s_t)$     // Select action according to the current policy
      $s_{t+1} \sim p(s_{t+1}|s_t, a_t), r_t$     // Observe reward $r_t$ and new state $s_{t+1}$
      $\mathcal{D} \leftarrow \mathcal{D} \cup \{(s_t, a_t, r_t, s_{t+1})\}$     // Store the transition in the replay buffer
   **end for**
   **for** each gradient step **do**
      Sample mini-batch $d_{trn}$ from $\mathcal{D}$
      Update $\theta \leftarrow Eq.$ (4)     // Update the critic parameters
      **meta-train:**
      $L^{\text{critic}} \leftarrow Eqs.$ (1), (2) $or$ (3)     // Vanilla-critic-provided loss for actor
      $L_\omega^{\text{mcritic}} \leftarrow Eqs.$ (10) $or$ (11)     // Meta-critic-provided loss for actor
      $\phi_{old} = \phi - \eta\nabla_\phi L^{\text{critic}}$     // Update actor according to $L^{\text{critic}}$ only
      $\phi_{new} = \phi_{old} - \eta\nabla_\phi L_\omega^{\text{mcritic}}$     // Update actor according to $L^{\text{critic}}$ and $L_\omega^{\text{mcritic}}$
      **meta-test:**
      Sample mini-batch $d_{val}$ from $\mathcal{D}$
      $L^{\text{meta}}(d_{val}; \phi_{new})$ $or$ $L_{clip}^{\text{meta}}(d_{val}; \phi_{old}, \phi_{new}) \leftarrow Eqs.$ (8) $or$ (9)     // Meta-loss
      **meta-optimisation**
      $\phi \leftarrow \phi - \eta(\nabla_\phi L^{\text{critic}} + \nabla_\phi L_\omega^{\text{mcritic}})$     // Update the actor parameters
      $\omega \leftarrow \omega - \eta\nabla_\omega L^{\text{meta}}$ $or$ $\omega - \eta\nabla_\omega L_{clip}^{\text{meta}}$     // Update the meta-critic parameters
   **end for**
**end for**=0

---

as measured by validation meta loss. The update of vanilla-critic is also in the lower loop, but as it updates as usual, we focus on the actor and meta-critic optimisation for simplicity of exposition.

In this setup the meta-critic is a neural network $h_\omega(d_{trn}; \phi)$ that takes as input some featurisation of the actor $\phi$ and the states and actions in $d_{trn}$. The meta-critic network must produce a scalar output, which we can then treat as a loss $L_\omega^{\text{mcritic}} := h_\omega$, and must be differentiable with respect to $\phi$. We next discuss the overall optimisation flow and the specific meta-critic architecture.

**Meta-Optimisation Flow.** To optimise Eq. (5), we iteratively update the meta-critic parameter $\omega$ (upper-level) and actor and vanilla-critic parameters $\phi$ and $\theta$ (lower-level). At each iteration, we perform: (i) Meta-train: Sample a mini-batch of transitions and putatively update policy $\phi$ based on the vanilla-critic-provided $L^{\text{critic}}$ and the meta-critic-provided $L_\omega^{\text{mcritic}}$ losses.
(ii) Meta-test: Sample another mini-batch of transitions to evaluate the performance of the updated policy according to $L^{\text{meta}}$. (iii) Meta-optimisation: Update meta-critic $\omega$ to maximize the performance on the validation batch, and perform the real actor update according to both losses. Thus the meta-critic co-evolves with the actor as they are trained online and in parallel. Figure 1 and Alg. 1 summarize the process and the details of each step are explained next. Meta-critic can be flexibly integrated with any OffP-AC algorithms, and the further implementation details for DDPG, TD3 and SAC are in the supplementary material.

**Updating Actor Parameters ($\phi$).** During meta-train, we sample a mini-batch of transitions $d_{trn} = \{(s_i, a_i, r_i, s_{i+1})\}$ with batch size $N$ from the replay buffer $\mathcal{D}$. We update the policy using both losses as:

$$\phi_{new} = \phi - \eta\frac{\partial\, L^{\text{critic}}(d_{trn})}{\partial\phi} - \eta\frac{\partial\, L_\omega^{\text{mcritic}}(d_{trn})}{\partial\phi}. \quad (6)$$

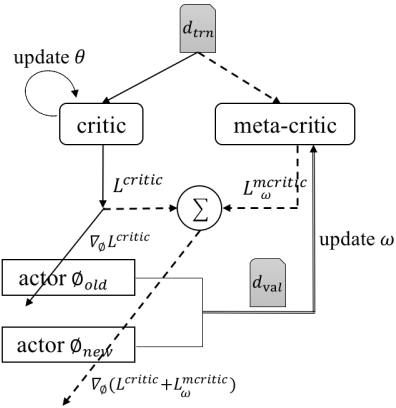

Figure 1: Meta-critic for OffP-AC. Train and validation data are sampled from the replay buffer during meta-train and meta-test. Actor parameters are updated based on vanilla-critic- and meta-critic-provided losses. Meta-critic parameters are updated by the meta-loss.

We also compute a separate update:

$$\phi_{old} = \phi - \eta \frac{\partial L^{\text{critic}}(d_{trn})}{\partial \phi} \tag{7}$$

that only leverages the vanilla-critic-provided loss. If meta-critic provided a beneficial source of loss, $\phi_{new}$ should be a better parameter than $\phi$, and in particular a better parameter than $\phi_{old}$. We will use this comparison in the next meta-test step.

**Updating Meta-Critic Parameters ($\omega$).** To train the meta-critic, we sample another mini-batch of transitions: $d_{val} = \{(s_i^{\text{val}}, a_i^{\text{val}}, r_i^{\text{val}}, s_{i+1}^{\text{val}})\}$ with batch size $M$. The use of a validation batch for bi-level meta-optimisation [8, 29] ensures the meta-learned component does not overfit. As our framework is off-policy, this does not incur any sample efficiency cost. The meta-critic is then updated by a meta-loss $\omega \leftarrow \omega - \eta L^{\text{meta}}(\cdot)$ that measures actor performance after learning.

**Meta-Loss Definition.** The most intuitive meta-loss definition is the validation performance of updated actor $\phi_{new}$ as measured by the normal critic:

$$L^{\text{meta}} = L^{\text{critic}}(d_{val}; \phi_{new}). \tag{8}$$

However, we find it helpful for optimisation efficiency and stability to optimise the clipped difference between updates with- and without meta-critic's input as:

$$L_{clip}^{\text{meta}} = \tanh(L^{\text{critic}}(d_{val}; \phi_{new}) - L^{\text{critic}}(d_{val}; \phi_{old})). \tag{9}$$

This is simply a monotonic re-centering and re-scaling of $L^{\text{critic}}$. (The parameter $\omega$ that minimizes $L_{clip}^{\text{meta}}$ as Eq. (9) also minimizes $L^{\text{meta}}$ of Eq. (8) and vice-versa.) Note that in Eq. (9) the updated actor $\phi_{new}$ depends on the feedback given by meta-critic $\omega$ and $\phi_{old}$ does not. Thus only the first term is optimised for $\omega$. In this setup the $L^{\text{critic}}(d_{val}; \phi_{new})$ term should obtain high reward/low loss on the validation batch and the latter $L^{\text{critic}}(d_{val}; \phi_{old})$ provides a *baseline*, analogous to the baseline widely used to accelerate and stabilize the policy-gradient RL. $\tanh$ ensures meta-loss range is always nicely distributed in $(-1, 1)$, and caps the magnitude of the meta-gradient. In essence, meta-loss is for the agent to ask itself: "Did meta-critic learning improve validation performance compared to vanilla learning?", and adjusts meta-critic $\omega$ accordingly. We will compare the options $L^{\text{meta}}$ and $L_{clip}^{\text{meta}}$ later.

**Designing Meta-Critic ($h_{\omega}$).** The meta-critic $h_{\omega}$ implements the additional loss for actor. The design-space for $h_{\omega}$ has several requirements: (i) Its input must depend on the policy parameters $\phi$, because this meta-critic-provided loss is also used to update the policy. (ii) It should be permutation invariant to transitions in $d_{trn}$, i.e., it should not make a difference if we feed the randomly sampled transitions indexed [1,2,3] or [3,2,1]. A naivest way to achieve (i) is given in MetaReg [2] which meta-learns a parameter regularizer: $h_{\omega}(\phi) = \sum_i \omega_i |\phi_i|$. Although this form of $h_{\omega}$ acts directly on $\phi$, it does not exploit state information, and introduces a large number of parameters in $h_{\omega}$, as $\phi$ may be a high-dimensional neural network. Therefore, we design a more efficient and effective form of $h_{\omega}$ that also meets both of these requirements. Similar to the feature extractor in supervised learning, the actor needs to analyse and extract information from states for decision-making. We assume the policy network can be represented as $\pi_{\phi}(s) = \hat{\pi}(\bar{\pi}(s))$ and decomposed into the feature extraction $\bar{\pi}_{\phi}$ and decision-making $\hat{\pi}_{\phi}$ (i.e., the last layer of the full policy network) modules. Thus the output of the penultimate layer of full policy network is just the output of feature extraction $\bar{\pi}_{\phi}(s)$, and such output of feature jointly encodes $\phi$ and $s$. Given this encoding, we implement $h_w(d_{trn}; \phi)$ as a three-layer multi-layer perceptron $f_{\omega}$ whose input is the extracted feature from $\bar{\pi}_{\phi}(s)$. Here we consider two designs for meta-critic ($h_{\omega}$): using our joint feature alone (Eq. (10)) or augmenting the joint feature with states and actions (Eq. (11)):

$$h_w(d_{trn}; \phi) = \frac{1}{N} \sum_{i=1}^{N} f_{\omega}(\bar{\pi}_{\phi}(s_i)), \qquad (10) \quad h_w(d_{trn}; \phi) = \frac{1}{N} \sum_{i=1}^{N} f_{\omega}(\bar{\pi}_{\phi}(s_i), s_i, a_i). \quad (11)$$

$h_{\omega}$ provides as an auxiliary critic whose input is based on the batch-wise set-embedding [43] of our joint actor-state feature. That is to say, $d_{trn}$ is a randomly sampled mini-batch transitions from the replay buffer, and then $s$ (and $a$) of transitions are inputted to $h_{\omega}$, and finally we obtain the meta-critic-provided loss for $d_{trn}$. Here, our design of Eq. (11) also includes the cues in LIRPG and EPG where $s_i$ and $a_i$ are used as the input of their learned reward and loss respectively. We set a softplus activation to the final layer of $h_{\omega}$, following the idea in TD3 that vanilla critic may over-estimate and so the a non-negative additional actor loss can mitigate such over-estimation. Moreover, note that only $s_i$ (and $a_i$) from $d_{trn}$ are used to calculate $L^{\text{critic}}$ and $L_{\omega}^{\text{mcritic}}$, while $s_i$, $a_i$, $r_i$ and $s_{i+1}$ are all used for optimising the vanilla critic.

# 4  Experiments and Evaluation

We take the algorithms DDPG, TD3 and SAC as our vanilla baselines, and denote their enhancements by meta-critic as DDPG-MC, TD3-MC, SAC-MC. All -MCs augment their built-in vanilla critic with the proposed meta-critic. We take Eq. (10) and $L_{clip}^{meta}$ as the default meta-critic setup, and compare alternatives in the ablation study. For our implementation of meta-critic, we use a three-layer neural network with an input dimension of $\bar{\pi}$ (300 in DDPG and TD3, 256 in SAC), two hidden feed-forward layers of 100 hidden nodes each, and ReLU non-linearity between layers.

**Implementation Details.** We evaluate the methods on a suite of seven MuJoCo tasks [39] in OpenAI Gym [4], two MuJoCo tasks in rllab [5], and a simulated racing car TORCS [22]. For MuJoCo-Gym, we use the latest V2 tasks instead of V1 used in TD3 and the old-SAC [12] without modification to their original environment or reward. We use the open-source implementations "OurDDPG"[2], TD3[3] and SAC[4]. Here, "OurDDPG" is the re-tuned version of DDPG implemented in Fujimoto et al. [9] with the same hyper-parameters. In MuJoCo cases we integrate our meta-critic with learning rate 0.001. The details of TORCS hyper-parameters are in the supplementary material. Our demo code can be viewed on `https://github.com/zwfightzw/Meta-Critic`.

## 4.1  Evaluation of Meta-Critic OffP-AC Learning

**DDPG.** Figure 2 shows the learning curves of DDPG and DDPG-MC. The results in each task are averaged over 5 random seeds (trials) and network initialisations. The standard deviation intervals are shown as shaded regions over time steps. Following Fujimoto et al. [9], curves are uniformly smoothed for clarity (window_size=10 for TORCS, 30 for others). We run MuJoCo-Gym tasks for 1-10 million depending on the environment, rllab tasks for 3 million and TORCS experiment for 100 thousand steps. Every 1000 steps we evaluate our policy over 10 episodes with no exploration noise. From Figure 2, DDPG-MC generally outperforms DDPG baseline in terms of the learning speed and asymptotic performance. Furthermore, -MC usually has smaller variance. The summary results for all tasks using the vanilla baseline and -MCs in terms of max average return are shown in Table 1. -MC usually provides consistently higher max return. We select seven tasks for plotting. The other MuJoCo tasks "Reacher", "InvPend" and "InvDouPend" have reward upper bounds that all methods can reach quickly without obvious differences.

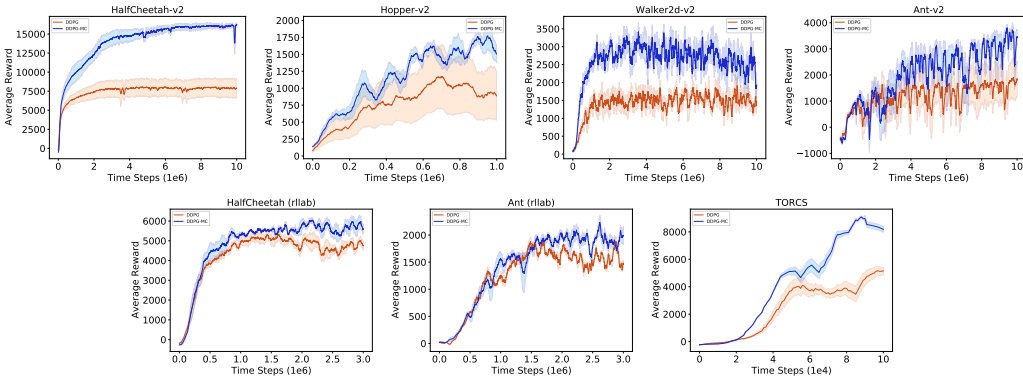

Figure 2: Learning curve Mean-STD of vanilla DDPG and DDPG-MC for continuous control tasks.

**TD3 and SAC.** Figure 3 reports the learning curves for TD3. For some tasks the vanilla TD3's performance declines in the long run, while TD3-MC shows improved stability with much higher asymptotic performance. Thus TD3-MC provides comparable or better learning performance in each case, while Table 1 shows the clear improvement in the max average return. For SAC in Figure 4, note that we use the most recent update of SAC [13], which is actually the combination of SAC+TD3. Although SAC+TD3 is arguably the strongest existing method, SAC-MC still gives a clear boost on the asymptotic performance for many tasks, especially the most challenging TORCS.

Table 1: Comparison of RL algorithms and their online meta-learning enhancements, including meta-learner PPO-LIRPG [44]. Max Average Return over 5 trials over all time steps. Max value for each comparison is in bold, and max value overall is underlined.

| Environment | DDPG | DDPG-MC | TD3 | TD3-MC | SAC | SAC-MC | PPO | PPO-LIRPG |
|---|---|---|---|---|---|---|---|---|
| HalfCheetah | 8440.2 | **15808.9** | 12735.7 | **15064.0** | 16651.8 | **16815.9** | 2061.5 | 1882.6 |
| Hopper | 1871.1 | **2776.7** | 3580.3 | **3670.4** | 3610.6 | **3738.4** | 3762.0 | 2750.0 |
| Walker2d | 2920.2 | **4543.5** | 5942.7 | **6298.0** | 6398.8 | **7164.9** | 4432.6 | 3652.9 |
| Ant | 2375.4 | **3661.1** | 5914.8 | **6280.0** | 6954.4 | **7204.3** | 684.2 | 23.6 |
| Reacher | **-3.6** | -3.7 | -3.0 | **-2.9** | -2.8 | **-2.7** | -6.08 | -7.53 |
| InvPend | **1000.0** | **1000.0** | **1000.0** | **1000.0** | **1000.0** | **1000.0** | 988.2 | 971.6 |
| InvDouPend | 9307.5 | **9326.5** | 9357.4 | **9358.8** | **9359.6** | **9359.6** | 7266.0 | 6974.9 |
| HalfCheetah(rllab) | 6245.6 | **7239.1** | 7648.2 | **8552.1** | 10011.0 | **10597.0** | - | - |
| Ant(rllab) | 2300.8 | **2929.4** | 3672.6 | **4776.8** | 8014.8 | **8353.8** | - | - |
| TORCS | 6188.1 | **9353.3** | 14841.7 | **33684.2** | 24674.7 | **32869.0** | - | - |

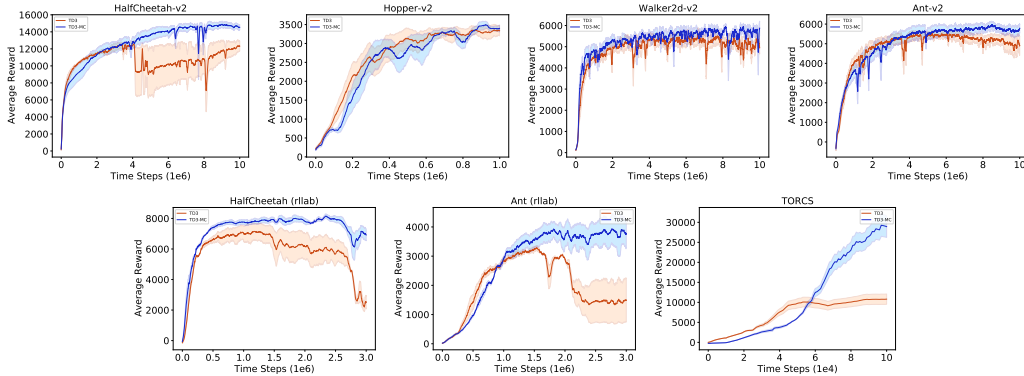

Figure 3: Learning curve Mean-STD of vanilla TD3 and TD3-MC for continuous control tasks.

**Comparison vs PPO-LIRPG.** Intrinsic Reward Learning for PPO [44] is the most related method to our work in performing online single-task meta-learning of an additional reward/loss. Their original PPO-LIRPG evaluated on a modified environment with hidden rewards. Here we apply it to the standard unmodified learning tasks that we aim to improve. Table 1 tells that: (i) In this conventional setting, PPO-LIRPG worsens rather than improves basic PPO performance. (ii) Overall OffP-AC methods generally perform better than on-policy PPO for most environments. This shows the importance of our meta-learning contribution to the off-policy setting. In general Meta-Critic is preferred compared to PPO-LIRPG because the latter only provides a scalar reward bonus that helps the policy indirectly via high-variance policy-gradient updates, while ours provides a direct loss.

**Summary.** Table 1 and Figure 5 summarize all the results by max average return. SAC-MC generally performs best and -MCs are generally comparable or better than their corresponding vanilla alternatives. -MCs usually provide improved variance in return compared to their baselines.

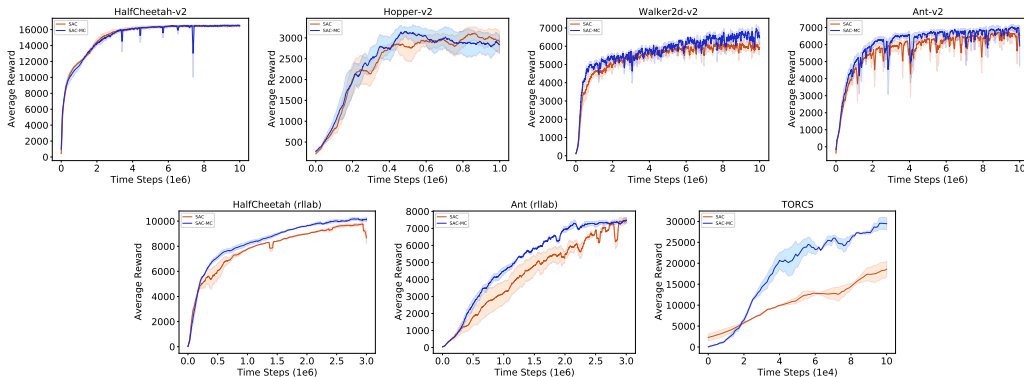

Figure 4: Learning curve Mean-STD of vanilla SAC and SAC-MC for continuous control tasks.

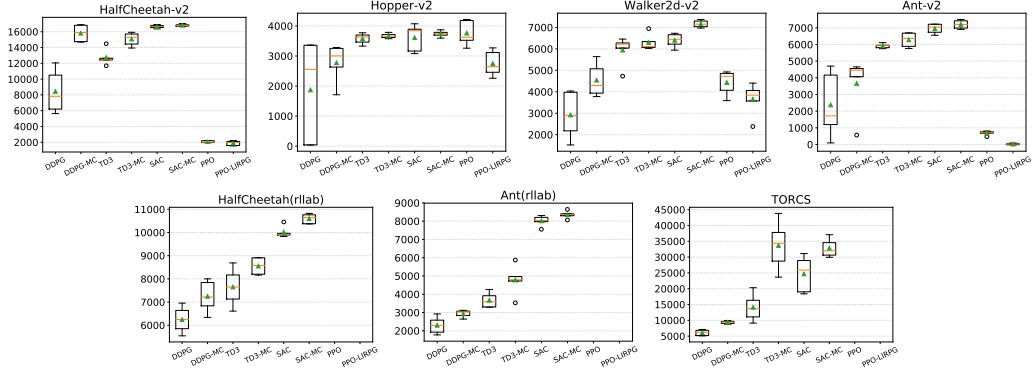

Figure 5: Comparison of RL algorithms and their online meta-learning enhancement. Box plots of the Max Average Return over 5 trials of all time steps.

## 4.2 Further Analysis

**Loss and Optimisation Analysis.** We take tabular MDP [6] ($|S| = 2$, $|A| = 2$) as an example using DDPG. Figure 6 first reports the normal $L^{\text{critic}}$ of actor, and the introduced $h_\omega$ (i.e., $L_\omega^{\text{mcritic}}$) and $L_{clip}^{\text{meta}}$ over 5 trials. We also plot model optimisation trajectories (pink dots) via a 2D weight-space slice in right part of Figure 6. They are plotted over the average reward surface. Following the network visualization in Li et al. [18], we calculate the subspace to plot as: Let $\phi_i$ denote model parameters at episode $i$ and the final estimate as $\phi_n$ (here $n = 100$). We apply PCA to matrix $M = [\phi_0 - \phi_n, \ldots, \phi_{n-1} - \phi_n]$, and take the two most explanatory directions of this optimisation path. Parameters are then projected onto the plane defined by these directions for plotting; and models at each point are densely evaluated to get average reward. Figure 6 shows: (i) DDPG-MC convergences faster to a lower value of $L^{\text{critic}}$, demonstrating the meta-critic's ability to accelerate learning. (ii) Meta-loss is randomly initialised at the start, but as $\omega$ begins to be trained via meta-test on validation data, meta-loss drops swiftly below zero and then $\phi_{new}$ is better than $\phi_{old}$. In the late stage, meta-loss goes towards zero, indicating all of $h_\omega$'s knowledge has been distilled to help the actor. Thus meta-critic is helpful in defining better update directions in the early stages of learning (but note that it can still impact later stage learning via changing choices made early). (iii) $L_\omega^{\text{mcritic}}$ converges smoothly under the supervision of meta-loss. (iv) DDPG-MC has a very direct and fast optimisation movement to the high reward zone of parameter space, while the vanilla DDPG moves slowly through the low reward space before finally finding the direction to the high-reward zone.

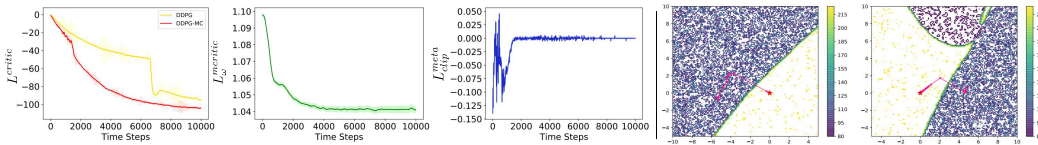

Figure 6: Left: Loss analysis of our algorithm. Right: Visualization of optimisation dynamics of vanilla DDPG (left) and DDPG-MC (right). The red star denotes the final optimisation point.

**Ablation on $h_\omega$ design.** We run Walker2d under SAC-MC with the alternative $h_\omega$ from Eq. (11) or in MetaReg [2] format (input actor parameters directly). In Table 2, we record the max average return and sum average return (area under the average reward curve) of evaluations over all time steps. Eq. (11) achieves the highest max average return and our default $h_\omega$ (Eq. (10)) attains the highest mean average return. We can also see some improvement for $h_\omega(\phi)$ in MetaReg format, but the huge number (73484) of parameters is expensive. Overall, all meta-critic designs provide at least a small improvement on vanilla SAC.

**Ablation on meta-loss design.** We considered two meta-loss designs in Eqs. (8&9). For $L_{clip}^{\text{meta}}$ in Eq. (9), we use $L^{\text{critic}}(d_{val}; \phi_{old})$ as a baseline to improve numerical stability of the gradient update. To evaluate this design, we also compare using vanilla $L^{\text{meta}}$ in Eq. (8). The last column in Table 2 shows vanilla $L^{\text{meta}}$ barely improves on vanilla SAC, validating our meta-loss design.

Table 2: Ablation study on different designs of $L_\omega^{\text{mcritic}}$ (i.e., $h_\omega$) and $L^{\text{meta}}$ applied to SAC-Walker2d. Max and Sum Average Return over 5 trials of all time steps. Max value in each row is in bold.

| | SAC | $L_{clip}^{\text{meta}} : \tanh(\phi_{new} - \phi_{old})$ | | | $L^{\text{meta}} : \phi_{new}$ |
| --- | --- | --- | --- | --- | --- |
| | | $h_\omega(\bar{\pi}_\phi)$ | $h_\omega(\bar{\pi}_\phi, s, a)$ | $h_\omega(\phi)$ | $h_\omega(\bar{\pi}_\phi)$ |
| Max Average Return | $6398.8 \pm 289.2$ | $7164.9 \pm 151.3$ | $\mathbf{7423.8 \pm 780.2}$ | $6644.3 \pm 1815.6$ | $6456.1 \pm 424.8$ |
| Sum Average Return | 53,695,678 | **61,672,039** | 57,364,405 | 58,875,184 | 52,446,717 |

**Controlling for compute cost and parameter count.** We find that meta-critic increases 15-30% compute cost and 10% parameter count above the baselines (the latter is neglectable as it is small compared to the replay buffer's memory footprint) during training, and this is primarily attributable to the cost of evaluating the meta-loss $L_{clip}^{\text{meta}}$ and hence $L_\omega^{\text{mcritic}}$. To investigate whether the benefit of meta-critic can be replicated simply by increasing compute expenditure or model size, we perform control experiments by increasing the vanilla baselines' compute budget or parameter count to match the -MCs. Specifically, if meta-critic takes $K\%$ more compute than the baseline, then we re-run the baseline with $K\%$ more update steps per iteration. This '+updates' condition provides the baseline with more mini-batch samples while controlling the number of environment interactions. Note that due to implementation constraints of SAC, increasing updates in 'SAC+updates' requires taking at least 2x gradient updates per environment step compared to SAC and SAC-MC. Thus it takes 100% more updates than SAC and significantly more compute time than SAC-MC. To control for parameter count, if meta-critic takes $N\%$ more parameters than baseline, then we increase the baselines' network size with $N\%$ more parameters by linearly scaling up the size of all hidden layers ('+params').

The max average return results for the seven tasks in these control experiments are shown in Table 3, and the detailed learning curves of the control experiments are in the supplementary material. Overall, there is no consistent benefit in providing the baseline with more compute iterations or parameters, and in many environments they perform worse than the baseline or even fail entirely, especially in '+updates' condition. Thus -MCs' good performance can not be simply replicated by a corresponding increase in gradient steps or parameter size taken by the baseline.

Table 3: Controlling for compute cost and parameter count. Max Average Return over 5 trials over all time steps. Max value for each comparison is in bold.

| Environment | DDPG | DDPG +updates | DDPG +params | DDPG -MC | TD3 | TD3 +updates | TD3 +params | TD3 -MC | SAC | SAC +updates | SAC +params | SAC -MC |
| --- | --- | --- | --- | --- | --- | --- | --- | --- | --- | --- | --- | --- |
| HalfCheetah | 8440.2 | 15004.5 | **15153.9** | 15808.9 | 12735.7 | 11585.2 | 11980.7 | **15064.0** | 16651.8 | 16309.9 | 16339.3 | **16815.9** |
| Hopper | 1871.1 | 1753.0 | 2438.0 | **2776.7** | 3580.3 | 2903.1 | 1460.0 | **3670.4** | 3610.6 | 3510.8 | 3441.6 | **3738.4** |
| Walker2d | 2920.2 | 3826.5 | 2964.3 | **4543.5** | 5942.7 | 3414.0 | 3185.4 | **6298.0** | 6398.8 | 6363.7 | 6423.7 | **7164.9** |
| Ant | 2375.4 | 1504.2 | 3615.1 | **3661.1** | 5914.8 | 1262.3 | 984.7 | **6280.0** | 6954.4 | 6253.6 | 5999.1 | **7204.3** |
| HalfCheetah(rllab) | 6245.6 | 6526.0 | 6589.0 | **7239.1** | 7648.2 | 8021.8 | **9003.0** | 8552.1 | 10011.0 | 9008.3 | 9122.0 | **10597.0** |
| Ant(rllab) | 2300.8 | 2875.4 | 2763.7 | **2929.4** | 3672.6 | 2838.1 | 3714.3 | **4776.8** | 8014.8 | 6464.4 | 6868.8 | **8353.8** |
| TORCS | 6188.1 | 4932.5 | 8104.7 | **9353.3** | 14841.7 | 20473.2 | 27850.4 | **33684.2** | 24674.7 | 11946.7 | 24932.4 | **32869.0** |

**Discussion.** We introduce an auxiliary meta-critic that goes beyond the information available to vanilla critic to leverage measured actor *learning progress* (Eq. (9)). This is a generic module that can potentially improve any off-policy actor-critic derivative-based RL method for a minor overhead at train time, and no overhead at test time; and can be applied directly to single tasks without requiring task-families as per most other meta-RL methods [3, 7, 15, 30]. Our method is myopic, in that it uses a single inner (base) step per outer (meta) step. A longer horizon look-ahead may ultimately lead to superior performance. However, this incurs the cost of additional higher-order gradients and associated memory use, and risk of unstable high-variance gradients [21, 29]. New meta-optimizers [24] may ultimately enable these issues to be solved, but we leave this to future work.

# 5 Conclusion

We present Meta-Critic, a *derivative-based* auxiliary critic module for *off-policy* actor-critic reinforcement learning methods that can be meta-learned online during *single task* learning. The meta-critic is trained to provide an additional loss for the actor to assist the actor learning progress, and leads to long run performance gains in continuous control. This meta-critic module can be flexibly incorporated into various contemporary OffP-AC methods to boost performance. In future work, we plan to apply the meta-critic to conventional meta-learning with multi-task and multi-domain RL.

## Acknowledgements

This work was partially supported by the National Natural Science Foundation of China (No. 61751208) and the Advanced Research Program (No. 41412050202) and the Engineering and Physical Sciences Research Council of the UK (EPSRC) Grant number EP/S000631/1.

## Broader Impact

We introduced a framework for meta RL where learning is improved through the addition of an auxiliary meta-critic which is trained online to maximise learning progress. This technology could benefit all current and potential future downstream applications of reinforcement learning, where learning speed and/or asymptotic performance can still be improved – such as in game playing agents and robot control.

Faster reinforcement learning algorithms such as meta-critic could help to reduce the energy requirements training agents, which can add up to a significant environmental cost [35]; and bring us one step closer to enabling learning-based control of physical robots, which is currently rare due to the sample inefficiency of RL algorithms in comparison to the limited robustness of real robots to physical wear and tear of prolonged operation. Returning to our specific algorithmic contribution, introducing learnable reward functions rather than relying solely on manually specified rewards introduces a certain additional level of complexity and associated risk above that of conventional reinforcement learning. If the agent participates in defining its own reward, one might like to be able to interpret the learned reward function and validate that it is reasonable and will not lead to the robot learning to perform undesirable behaviours. This suggests that development of explainable AI techniques suited for reward function analysis could be a good topic for future research.

## Footnotes

[1]See Franceschi et al. [8] for a discussion on convergence of bi-level algorithms.

[2]https://github.com/sfujim/TD3/blob/master/OurDDPG.py

[3]https://github.com/sfujim/TD3/blob/master/TD3.py

[4]https://github.com/pranz24/pytorch-soft-actor-critic

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
