[Supplementary Material]

# Supplementary Material

## 1 Algorithms of Meta-Critic for DDPG, TD3 and SAC

We incorporate our Meta-Critic to the implementation of vanilla DDPG, TD3 and SAC, following their original implementations. The implementations are summarised by the pseudocode in Algorithms 1-3.

---

**Algorithm 1** DDPG-MC algorithm

---

Initialize critic $Q(s, a|\theta)$, actor $\pi(s|\phi)$ and meta-critic network $h_\omega$
Initialize target network $Q'$ and $\pi'$ with weights $\theta' \leftarrow \theta$, $\phi' \leftarrow \phi$
Initialize replay buffer $\mathcal{R}$
**for** episode = 1, ..., M **do**
  Initialize a random process $\mathcal{N}$ for action exploration
  Receive initial observation state $s_1$
  **for** t = 1, ..., T **do**
    Select action $a_t = \pi(s_t|\phi) + \mathcal{N}_t$ according to the current policy and exploration noise
    Execute action $a_t$, observe reward $r_t$ and new state $s_{t+1}$
    Store transition $(s_t, a_t, r_t, s_{t+1})$ in $\mathcal{R}$
    Sample a random mini-batch of $N$ transitions $(s_i, a_i, r_i, s_{i+1})$ from $\mathcal{R}$
    Set $y_i = r_i + \gamma Q'(s_{i+1}, \pi'(s_{i+1}|\phi')|\theta')$
    Update critic by minimizing the loss: $L = N^{-1} \sum_i (y_i - Q(s_i, a_i|\theta))^2$
    **meta-train**:
    Calculate the old actor weights using the normal actor loss from vanilla critic:

$$\nabla_\phi L^{\text{critic}} = -N^{-1} \sum_i \nabla_a Q(s, a|\theta)|_{s=s_i, a=\pi(s)} \nabla_\phi \pi(s|\phi)|_{s=s_i}$$

$$\phi_{old} = \phi - \eta \nabla_\phi L^{\text{critic}}$$

    Calculate the new actor weights using the additional actor loss from meta-critic:

$$\nabla_\phi L_\omega^{\text{mcritic}} = \nabla_\phi h_\omega = N^{-1} \sum_i \nabla_\phi f_\omega(\bar{\pi}(s|\phi)|_{s=s_i})$$

$$\phi_{new} = \phi_{old} - \eta \nabla_\phi L_\omega^{\text{mcritic}}$$

    **meta-test**:
    Sample a random mini-batch of $N$ $s_i^{\text{val}}$ from $\mathcal{R}$
    Calculate the meta-loss using the meta-test sampled transitions:

$$L_{clip}^{\text{meta}} = \tanh(L^{\text{critic}}(s, a|\theta)|_{s=s_i^{\text{val}}, a=\pi(s|\phi_{new})} - L^{\text{critic}}(s, a|\theta)|_{s=s_i^{\text{val}}, a=\pi(s|\phi_{old})})$$

    **meta-optimisation**: Update the weight of actor and meta-critic network:

$$\phi \leftarrow \phi - \eta(\nabla_\phi L^{\text{critic}} + \nabla_\phi L_\omega^{\text{mcritic}})$$

$$\omega \leftarrow \omega - \eta \nabla_\omega L_{clip}^{\text{meta}}$$

    Update the target networks:

$$\theta' \leftarrow \tau\theta + (1 - \tau)\theta'$$

$$\phi' \leftarrow \tau\phi + (1 - \tau)\phi'$$

  **end for**
**end for**=0

---

---

**Algorithm 2** TD3-MC algorithm

---

Initialize critics $Q_{\theta_1}$, $Q_{\theta_2}$, actor $\pi_\phi$ and meta-critic network $h_\omega$

Initialize target networks $\theta'_1 \leftarrow \theta_1$, $\theta'_2 \leftarrow \theta_2$, $\phi' \leftarrow \phi$

Initialize replay buffer $\mathcal{B}$

**for** $t = 1$ **to** $T$ **do**

    Select action with exploration noise $a \sim \pi_\phi(s) + \epsilon$, $\epsilon \sim \mathcal{N}(0, \sigma)$ and observe reward $r$ and new state $s'$

    Store transition tuple $(s, a, r, s')$ in $\mathcal{B}$

    Sample mini-batch of $N$ transitions $(s, a, r, s')$ from $\mathcal{B}$

    $\tilde{a} \leftarrow \pi_{\phi'}(s') + \epsilon$,    $\epsilon \sim clip(\mathcal{N}(0, \tilde{\sigma}), -c, c)$

    $y \leftarrow r + \gamma \min_{i=1,2} Q_{\theta'_i}(s', \tilde{a})$

    Update critics $\theta_i \leftarrow \arg\min_{\theta_i} N^{-1} \sum (y - Q_{\theta_i}(s, a))^2$

    **if** $t \bmod d$ **then**

        **meta-train** :

        $\nabla_\phi L^{\text{critic}} = -N^{-1} \sum \nabla_a Q_{\theta_1}(s, a)|_{a=\pi_\phi(s)} \nabla_\phi \pi_\phi(s)$

        $\nabla_\phi L_\omega^{\text{mcritic}} = \nabla_\phi h_\omega = N^{-1} \sum \nabla_\phi f_\omega(\bar{\pi}_\phi(s))$

        Calculate the old actor weights using the normal actor loss: $\phi_{old} = \phi - \eta \nabla_\phi L^{\text{critic}}$

        Calculate the new actor weights using the additional actor loss: $\phi_{new} = \phi_{old} - \eta \nabla_\phi L_\omega^{\text{mcritic}}$

        **meta-test**:

        Sample mini-batch of $N$ $s^{\text{val}}$ from $\mathcal{B}$

        Calculate the meta-loss using the meta-test sampled transitions:

        $L_{clip}^{\text{meta}} = \tanh(L^{\text{critic}}(s^{\text{val}}, a|\theta_1)|_{a=\pi(s^{\text{val}})|\phi_{new}} - L^{\text{critic}}(s^{\text{val}}, a|\theta_1)|_{a=\pi(s^{\text{val}})|\phi_{old}})$

        **meta-optimisation**:

        Update the actor and meta-critic:

        $\phi \leftarrow \phi - \eta(\nabla_\phi L^{\text{critic}} + \nabla_\phi L_\omega^{\text{mcritic}})$

        $\omega \leftarrow \omega - \eta \nabla_\omega L_{clip}^{\text{meta}}$

        Update target networks:

        $\theta'_i \leftarrow \tau \theta_i + (1 - \tau)\theta'_i$

        $\phi' \leftarrow \tau \phi + (1 - \tau)\phi'$

    **end if**

**end for**=0

---

---

**Algorithm 3** SAC-MC algorithm

---

$\theta_1, \theta_2, \phi, \omega$      // Initialize parameters
$\bar{\theta} \leftarrow \theta_1, \bar{\theta}_2 \leftarrow \theta_2$      // Initialize target network weights
$\mathcal{D} \leftarrow \emptyset$      // Initialize an empty replay pool
**for** each iteration **do**
    **for** each environment step **do**
        $a_t \sim \pi_\phi(a_t|s_t)$      // Sample action from the policy
        $s_{t+1} \sim p(s_{t+1}|s_t, a_t)$      // Sample transition from the environment
        $\mathcal{D} \leftarrow \mathcal{D} \cup \{(s_t, a_t, r(s_t, a_t), s_{t+1})\}$      // Store the transition in the replay pool
    **end for**
    **for** each gradient step **do**
        $\theta_i \leftarrow \theta_i - \lambda_Q \nabla_{\theta i} J_Q(\theta_i)$ for $i \in \{1, 2\}$      // Update the Q-function parameters
        **meta-train** :
        $\nabla_\phi L^{\text{critic}} = N^{-1} \sum_t \nabla_a [\alpha \log (\pi_\phi(a|s)) - Q_\theta(s, a)|_{s=s_t, a=\pi(s)}] \nabla_\phi \pi_\phi(s)|_{s=s_t}$
        $\phi_{old} = \phi - \eta \nabla_\phi L^{\text{critic}}$      // Calculate old weights of the actor
        $\nabla_\phi L_\omega^{\text{mcritic}} = \nabla_\phi h_\omega = N^{-1} \sum_t \nabla_\phi f_\omega(\bar{\pi}_\phi(s))|_{s=s_t}$
        $\phi_{new} = \phi_{old} - \eta \nabla_\phi L_\omega^{\text{mcritic}}$      // Calculate new weights of the actor
        **meta-test**:
        $L_{clip}^{\text{meta}} = \tanh(L^{\text{critic}}(s, a|\theta)|_{s=s_t^{\text{val}}, a=\pi(s|\phi_{new})} - L^{\text{critic}}(s, a|\theta)|_{s=s_t^{\text{val}}, a=\pi(s|\phi_{old})})$
              // Calculate meta-loss

        **meta-optimisation**:
        $\phi \leftarrow \phi - \eta(\nabla_\phi L^{\text{critic}} + \nabla_\phi L_\omega^{\text{mcritic}})$      // Update the actor parameters
        $\omega \leftarrow \omega - \eta \nabla_\omega L_{clip}^{\text{meta}}$      // Update the meta-critic parameters

        $\alpha \leftarrow \alpha - \lambda \nabla_\alpha J(\alpha)$      // Adjust temperature
        $\bar{\theta}_i \leftarrow \tau \theta_i + (1 - \tau)\bar{\theta}_i$ for $i \in \{1, 2\}$      // Update target network weights
    **end for**
**end for** =0

---

## 2 Average Rewards on Other Tasks and PPO-LIRPG Experiments

This section includes additional results for the Reacher, Inverted Pendulum, and Inverted Double Pendulum environments. These were omitted from the main paper due to being relatively easy for all methods to saturate the maximum reward, as shown in Figures 1-3 and Figure 4.

Figure 1: Learning curve Mean-STD of vanilla DDPG and DDPG-MC for MuJoCo tasks with upper reward bound.

Figure 2: Learning curve Mean-STD of vanilla TD3 and TD3-MC for MuJoCo tasks with upper reward bound.

Figure 3: Learning curve Mean-STD of vanilla SAC and SAC-MC for MuJoCo tasks with upper reward bound.

Figure 4: Box plots of the Max Average Return over 5 trials of all time steps for MuJoCo tasks with upper reward bound.

Figure 5: Learning curve Mean-STD of PPO and PPO-LIRPG for continuous control tasks.

# 3 Hyper-parameters for TORCS

In DDPG and TD3 experiments, actor_lr=0.0003, critic_lr=0.001; for DDPG-MC and TD3-MC, omega_lr=0.0001.

In SAC experiments, hyper-parameters for TORCS are the same as for MuJoCo: actor_lr=0.0003, critic_lr=0.0003; for SAC-MC, omega_lr=0.001.

# 4 Control Experiments on Compute and Parameters

The detailed learning curves of the vanilla baselines, '-MC', '+updates' and '+params' variants under DDPG, TD3 and SAC are shown in Figure 6, 7 and 8 respectively. The effect of compute and parameter increased control baselines varies by environment and base algorithm, sometimes improving performance but often significantly worsening it. Unlike meta-critic, they do not show a consistent improvement on the vanilla methods.

Figure 6: Learning curve Mean-STD of DDPG-MC and baseline variants for continuous control tasks.

Figure 7: Learning curve Mean-STD of TD3-MC and baseline variants for continuous control tasks.

Figure 8: Learning curve Mean-STD of SAC-MC and baseline variants for continuous control tasks.