[Reviews · NeurIPS 2020]

Review 1

Summary and Contributions: This paper introduces a meta-learning method that improves performance of actor-critic reinforcement learning algorithms. The authors achieve that by adding a meta-learning loss to the critic, which in combination with the standard TD-error improves the performance of the actor. The method is general enough to be applied to any actor-critic RL algorithm. ------- after rebuttal -------- Given the authors' response I'm willing to keep my current, positive score but I really hope that the authors introduce all the promised changes.

Strengths: - The paper introduces a novel idea that I haven't seen before. Meta-RL has been traditionally applied to multi-task settings whereas the authors show how it can be done in a single-task actor-critic scenario. - The authors provide very interesting analysis in Fig. 5 indicating on a simple example, why and how their method works - The authors provide interesting ablation experiments explaining why they made certain decisions

Weaknesses: - Supplementary material is very valuable and provides much more context than what we see in the paper in terms of the final results. Unfortunately, the results presented in the paper don't normalize the aspects of the environments that vary across different methods, making it very difficult to compare the algorithms - In terms of the supplementary material, the authors only show the parameter, and updates-adjusted results on 1-2 environments and they don't comment on other environments. These are the most important results in the paper! This makes it impossible to judge the actual contribution of their algorithm

Correctness: - The method is correct and generally makes sense. I particularly appreciate the authors' effort to explain and showcase their method in a toy setting in Fig 5. - My main criticism is about the empirical methodology in the main paper. The authors introduce additional complexity but also increase the number of parameters and gradient steps compared to the baselines. The results that take that into account are only in the supplementary section and they are cherry-picked (shown on an extremely small subset of algorithms and environments). These results should be presented across all the environments and basically replace the current results in the main paper.

Clarity: I have reviewed this paper before and it's disappointing to see that some of my previous remarks haven't been addressed. In particular, Sec. 3.1 provides background and review of off-policy algorithm, it has nothing to do with the method. I'd really encourage the authors to organize the paper better and provide the necessary RL background in the background section. Line 200 also belongs to the background, it doesn't add anything to the method and it describes already well-established algorithms. The current split between sections makes it difficult to separate the authors' contributions and prior work. Discussion in 284 is quite unclear, especially line 286, given that it the given intuition wasn't explained earlier in the paper. Please provide more context for this interpretation.

Relation to Prior Work: Yes.

Reproducibility: Yes

Additional Feedback: Please provide more comprehensive results on parameter and updates-adjusted version of the baselines. Otherwise, it's very difficult to judge the contribution of your algorithm.


Review 2

Summary and Contributions: This paper proposes a method to improve final performance for off-policy actor-critic methods, by adding a learned loss to the critic that is meta-learned online alongside the off-policy method on a single task. In the paper, the authors give an overview of their algorithm and then dive into the different aspects of it in more detail. Empirically, the proposed meta-critic is applied on top of three off-policy methods (DDPG, TD3, SAC) on seven different environments (MuJoCo tasks from gym and rllab, and a simulated racing car environment called TORCS) for five seeds each. For DDPG, the performance increase is significant on most environments, for TD3 and SAC the performance is either similar or slightly higher (except on TORCS where the performance increase is pretty significant). ----------------------------------------------------------------------- ----------------------------------------------------------------------- Update: I read the other reviews and the authors feedback. R2Q1 - Thanks for the response, it would be great if you could include such a discussion in a revised version of the paper. Additional experiments / comparisons to other auxiliary tasks that boost performance could strengthen the paper even further IMO but are not strictly necessary. R2Q2 - Thanks. Again, bringing this up in the paper even in 1-2 sentences would be nice for the reader just to be aware of this. I still wonder if there's situations where a non-myopic loss is necessary and the current approach would break down, and am curious to potentially read about this in future work. Given the author's feedback and promises made for updating the paper I'm keeping my current score.

Strengths: The proposed method is well explained, and is general in that it can be applied to any off-policy algorithms. I like that the authors outline how to exactly do this for three prominent off-policy actor-critic methods, and explain how to add the meta-critic loss for each. The empirical results on DDPG are strong compared to the baselines.

Weaknesses: The value of the proposed method for me is twofold: First, it is interesting to see that meta-learning a critic loss online alongside an RL algorithm works at all, and is an interesting addition to the literature that looks at improving performance for *single* tasks using meta-learning. Second, this method could be used to increase performance when training agents on an environment and where we care about getting a high end-performance. A weakness that I see in the paper is that I find it hard to assess how feasible / useful this actually is. Why and when should I use the meta-critic on top of my off-policy algorithm, compared to other options? How does this compare to other auxiliary losses that can just be added on top of off-policy actor-critic algorithms? (Which ones are there?) How much does it increase my computation time / memory footprint? (Some information on this is in the appendix; I think 1-2 sentences about this in the main text would be nice as well.) So for me, any information that the authors can include into the paper that might help somebody make a decision about whether/when/where to use meta-critic would strengthen the paper.

Correctness: Yes

Clarity: The paper is overall well written. The motivation and problem setting of the paper are clearly stated and explained, and the results well explained. A few small things that should be clarified in the paper (not necessary in rebuttal): • How is the reward function defined? In some MuJoCo environments it also depends on the next state, which means H (line 100) should include s_{H+1}. • Line 102: "policy \phi" -> "policy \pi_\phi" • Line 225, "standard deviation confidence intervals" - please clarify in the paper. Do you show one standard deviation, or do you show (X%?) confidence intervals? • Algorithm 1, line 4+5 in "meta-train" part: Shouldn't it be \eta instead of \alpha?

Relation to Prior Work: Yes, but see comments above about work on auxiliary losses that can be added on top of the normal RL loss. It might be worth adding [1] and [2] to the related work section as well. [1] "Meta-Learning via Learned Loss" - Sarah Bechtle, Artem Molchanov, Yevgen Chebotar, Edward Grefenstette, Ludovic Righetti, Gaurav Sukhatme, Franziska Meier [2] "Improving generalization in meta reinforcement learning using learned objectives." - Louis Kirsch, Sjoerd van Steenkiste, Jürgen Schmidhuber. (cited already but only in introduction)

Reproducibility: Yes

Additional Feedback: An additional question to the authors: • The meta-loss is myopic, in the sense that it only looks at the improvement that the meta-critic loss gives for one gradient step. Do the authors think this could be a problem in the sense that it is too short-sighted? Would it be possible to do this for more than one gradient update step, and what would the trade-offs in terms of computation and memory overhead be?


Review 3

Summary and Contributions: In this paper, the authors focus on the actor-critic algorithms of off-policy reinforcement learning. Specifically, the authors propose a method of meta-learning an auxiliary critic function to help improve the actor. The auxiliary critic function is learned with a meta-optimization objective such that the actor achieves low loss on the normal critic after being trained with the auxiliary critic. The policy is then trained with both the normal critic loss and the auxiliary critic loss. The authors implement the proposed method on top of 3 commonly used actor-critic algorithms: DDPG, TD3 and SAC. The authors evaluate the proposed method in MuJoCo robot locomotion environments and a simulated racing car environment. The empirical evidence suggests that the proposed method achieves almost uniform improvements compared to the baselines.

Strengths: The empirical evaluation of the proposed method is really comprehensive. The authors evaluate the proposed meta-critic method on top of 3 common actor-critic algorithms and in a wide variety of environments. The results show almost uniform improvement on all base algorithms and environments. It can be clearly seen that the proposed method improves the sample efficiency for actor-critic methods. The proposed framework for meta-learning an auxiliary critic is general and therefore is applicable to many variants of actor-critic algorithms. Also more flexible designs of the auxiliary critic could be applied to obtain better performance.

Weaknesses: I am not very convinced that it is fair to compare the proposed method to vanilla actor critic algorithms (DDPG, TD3 and SAC) in their default configuration. With the proposed method, at every gradient step, two different batches of data are used to train the meta-critic. This means that the meta-training process for the proposed method is implicitly taking more gradient steps compared to the vanilla actor critic algorithm. The paper does include experiments in the appendix about taking more gradient steps by directly scaling the number up proportional to the computation. While this is an important comparison, it would also be important to evaluate the baselines with different number of gradient steps per environment step (ranging from 2 to 8). This is because taking more gradient steps per environment step could speed up training but taking too many steps would result in overfit. The design for the meta-critic architecture seems a little arbitrary. While the authors include an ablation study comparing two designs and the meta-regularization, it would be great to include experiments with other types of meta-critic designs, such as one taking policy actions, Q values and next states.

Correctness: The proposed claims and evaluation methodologies are correct to my knowledge.

Clarity: The paper is very well written. The proposed method and experiments are easy to interpret.

Relation to Prior Work: The relation to prior work is clearly discussed in this paper.

Reproducibility: Yes

Additional Feedback: ### Response to Author Feedback ### I went though the authors feedback and I don't think it has sufficiently addressed my concerns. For the gradient steps, while the policy does not receive an extra batch of data, it could potentially benefit from a better critic which does receive more data. Hence. I still think that it is important to compare to the baseline algorithms with 2 gradient steps at least. I will keep my score for this paper.


Review 4

Summary and Contributions: In this paper, the authors propose a meta critic to use in meta-reinforcement learning.

Strengths: I find the idea of using a validation set of transitions a sound idea. The idea seems simple, novel and significant. The reviewer is also glad to see that the authors have tested their idea on multiple algorithms and have shown that the idea works (or does not degrade performance), regardless of whether it is used on top of DDPG, TD3, SAC or PPO. This is the main reason I would recommend on accepting the paper, as it indicates significance and relevance to a broader range of algorithms.

Weaknesses: I find the theoretical grounding light, but it did not bother me. I think the bigger weakness is the evaluation in section 4.1 and Figure 2. Those curves look weird. I.e. on Figure 2 for Ant rllab, around 2.5e6 steps, all 5 seeds seem to drop simultaneously (the uncertainty bound is very small throughout the run, yet the mean is noisy). I would look for a bug in the setup, or at least say that the results are presented in a misleading manner. Additionally, in Figure 4 for the Torcs task, all seeds for SAC seemed to decay in the same way at the same time. I am skeptical of those results, but presume they are due to a bug in the way the figure was made, and not in the setup.

Correctness: The methodology is most likely correct, but I find the figures used misleading. The uncertainty bounds depicted are most likely not catching the true variation across seeds and steps. EDIT: on the figures: there might indeed have gone something wrong with the smoothing, where the smoothing accidentally happened across seeds, thus removing most of the variation between seeds in the final plots. I have compared with the TD3 paper, but in that paper the uncertainty bounds do seem plausible. I would like the issue to be solved in the final version. Now we finally have multiple seeds in reinforcement learning papers, it would be sad if the variation is not actually making it into the final paper. And thank you for pointing out the meta-gradient work, I guess the title does seem more promising when coming from that angle.

Clarity: The paper is reasonably clearly written, with a clear structure throughout. A bit more proof-reading would be welcome. Small nits: sort references on line 28, line 228: "on to environment" instead of "on the environment". Line 191 "embdedding". I find the connections being made to meta-learning tenuous, and the paper would be more clear if they were left out, in my opinion. I had a confusing first read through of the paper as I was expecting a meta-learning paper, quod non in my opinion.

Relation to Prior Work: Yes. I am not aware of previous work which was should have been mentioned.

Reproducibility: Yes

Additional Feedback:

[Author Response · NeurIPS 2020]

**R1Q1:** *Results didn't normalize .... across different methods.* **R1Q2:** *Only show params and updates-adjusted*
*results on several envs in supplementary...* **A:** Thanks for the suggestion. We'll re-organize the paper and present
all results controlling for parameter count and gradient-steps. We are conducting the rest of these experiments now.
From the results, we can see that there is no consistent improvement in performance for the baselines, and MC's good
performance cannot be replicated simply by a corresponding increase in params and update steps of the baseline.

**R1Q3:** *Organize the paper better.* **A:** A great suggestion. We understand your concerns clearly this time. We have
moved Sec 3.1 and Line 200 to a re-organized RL preliminaries section. It will be much clearer in the final paper.

**R1Q4:** *Discussion is unclear.* **A:** This discussion attempts to give a high-level intuition for how MC could benefit the
vanilla actor-critic (AC) baseline. MC optimizes learning progress (Eq 5) as measured by a 'validation' set (off-policy).
A conjecture about how this could improve conventional AC return maximisation is to increase visits to states with low
episodic-return (contrary to vanilla AC), but which are informative for *learning* (Eq 5), and thus longer term return.

**R2Q1:** *Whether/when/where to use meta-critic?* **A:** Thanks. (1) Most fundamentally, MC is relevant to *off-policy*,
*single-task*, *derivative-based* RL. If on-policy or evolutionary learning is desired, or the application is multi-task, then
other meta-RL methods are more suitable. (2) MC can potentially be used with any OffP-AC method since results
show similar or better performance than several baselines. (3) Our cost is 15-30% compute and 10% parameter count
above the baselines (latter is neglectable as small compared to replay buffer) during training, with no overhead at
testing-time. (4) Alternative auxiliary losses span from low-cost entropy (already in SAC) to hand-crafted unsupervised
reconstruction losses e.g., (Jaderberg ICLR'17, 'Reinforcement learning with unsupervised auxiliary tasks') that are
primarily relevant for pixel inputs; and meta-learned (LIRPG [41]) which should impose comparable overhead to ours.

**R2Q2:** *The meta-loss is myopic. Is it shortsighted? Useful to look ahead more than one gradient update step?* **A:**
Thanks for the suggestion. We agree it is myopic, and using more than one gradient step is potentially valuable in
principle. However in practice this would require back-propagating through a longer inner loop, which raises several
challenges: (1) Additional higher-order gradient calculation, and associated memory use. (2) Risk of vanishing or
unstable high-variance gradients. (Challenges are as discussed in other papers (iMAML NeurIPS'19, Taming-MAML
ICML'19).) Some other meta-RL studies consider longer episode length such as EPG (Houthooft NeurIPS'18), but
use zero-order optimization and on-policy learning. Nevertheless it is significant that we are able improve off-policy
learning with online meta-RL, even myopically. Designing an effective longer-horizon extension is left to future work.

**R2Q3:** *Related work and minor points.* **A:** Thanks. We will address these points.

**R3Q1:** *Fair comparison... number of gradient steps?* **A:** Thanks. Consider our policy ($\phi$) and auxiliary loss ($\omega$)
modules. The policy module always takes exactly as many gradient steps and data samples as the baselines. The loss
module takes an additional gradient step and sees an additional validation batch from the replay buffer. But this data is
not directly accessible to the policy module. So we do not see it as more data or more gradient steps for the policy per-se.
The experiment in the supplementary is motivated by controlling for total compute time. We agree that investigating
the impact of gradient-steps-per-environment-step is an interesting topic, but this is an orthogonal question getting out
of the scope of our work because (i) it is a hyperparameter of interest to study for all the base RL algorithms without
meta-learning, (ii) the same hyperparameter can be varied for both the baselines and the policy module in meta-critic.

**R3Q2:** *Arbitrary meta-critic loss design?* **A:** The meta-critic architecture $h_\omega$ is motivated by the need to input at
minimum parameters $\pi$ and states $s_i$, but $\pi$ is high dimensional (70k param), making $\omega$ easy to overfit if $[\pi, s_i]$ is input.
The trick of inputting $\bar{\pi}(s_i)$ means that both inputs are available but low-dimensional (300 param). Our alternatives
were partly inspired by the form of $v(s)$ and $q(s,a)$. Including the Q-value itself may suffer value estimation uncertainty
(e.g., overestimation). But we conducted this suggested experiment $h_\omega(\bar{\pi}(s_i), q(s_i, a_i), s_{i+1})$ as per Walker-2d/Table 2.
The Max Avg. Ret. is $5935.1 \pm 648.4$ and Sum Avg. Ret. is $52,098,042$. So including Q-value performs a bit worse.

**R4Q1:** *Some figures are misleading.* **A:** Thanks for the careful reading. We can confirm there are no code bugs. For
curves like ant-rllab, the concern may be because we smoothed the curve (using window_size=30, following TD3). In
addition, there's the STD but overlapped by the orange line. TORCS is not widely included in the benchmark suite of
existing algorithms, so they may be less well tuned for it in terms of stability (leading to mid-learning performance
drops). However, interestingly we did not have to tune MC to improve performance here. We'll resize smooth window,
adjust line color/transparency, and importantly include more seeds in the final paper to make the true variation clearer.

**R4Q2:** *Connections being made to meta-learning tenuous.* **A:** We presume the reviewer is taking meta-learning to refer
specifically to multi-task meta-learning as per MAML, RL2, PEARL, etc. We note that the term meta-learning is also
applied in single-task RL (when higher-order meta-gradients are used to train some aspect of the learning algorithm by
backprop through inner learning steps). For example [39], [13], (IJCAI'19, '*Meta-gradient Descent For Reinforcement*
*Learning Control*'), (ICML'18, '*Learning To Explore With Meta-Policy Gradient*'). IE: The commonality is the use of
meta-gradients, rather than the multi-task setting specifically. We use meta-gradients to train our auxiliary loss online.

[Meta-Review · NeurIPS 2020]

All of the reviewers maintained their score of a 6. Though, multiple reviewers considered lowering their score because they were disappointed by the author response and disheartened by the comments from R1 about how the authors did not address some of the comments from the previous round of reviewing. While this paper has flaws, particularly around the formulation and the experimental results, the paper may open up a new research direction of using meta-learned objectives to accelerate off-policy RL, a point that was particularly appreciated by R3. As such, I think that the paper should be accepted. Nonetheless, the authors are strongly encouraged to carefully read through each of the reviewer's comments (including the new comments in the updated reviews) and revise the paper to address the concerns.